# OpenReview forum: "Top of the CLASS: Benchmarking LLM Agents on Real-World Enterprise Tasks"
_ICLR.cc/2025/Workshop/BuildingTrust — BuildingTrust_

### Official Review · Reviewer_cBfm · 2025-02-23
**CLASSIC is a benchmark evaluating classification capabilities of agents triggering different workflows in enterprise domains.**

**Rating:** 6
**Confidence:** 5

**Review:**

**Strengths:**
- **Real-World Data:** Utilizes a dataset of 2,133 genuine user-chatbot conversations and 423 workflows from seven enterprise domains, offering a more realistic evaluation environment compared to synthetic benchmarks.
- **Evaluation:** Assesses LLMs on a broad range of metrics (Cost, Latency, Accuracy, Stability, and Security), going beyond just accuracy.

**Weaknesses:**
- **Task Specificity:** The benchmark focuses solely on a multiclass classification task (selecting the appropriate workflow), which may not capture the full spectrum of enterprise operational challenges. Many works have also found classification is largely solved with good in-domain data.
- **Cost Analysis Ambiguity:** The cost metric comparison could benefit from a more detailed breakdown to contextualize the trade-offs between different models. Agent trajectories could be useful to analyze.
- **Limited Multi-Turn Interaction:**
  - *"Most (85%) of the conversations in our dataset are single-turn dialogues."*
  - This leads the benchmark to not necessarily evaluate agents in the sense of multi-turn decision-making, but rather the classification capabilities of LLMs in enterprise workflows.
  - This distinction and the definition of "agent" being used should be more clearly articulated.

This is a useful benchmark for certain enterprise use cases but does not necessarily tread much new ground compared to previous agent benchmarks.

---

### Official Review · Reviewer_h5PR · 2025-03-02
**Good paper with some methodological limitations**

**Rating:** 7
**Confidence:** 2

**Review:**

This paper introduces CLASSIC, a benchmark evaluating LLM agents on enterprise tasks using actual chatbot interactions across cost, latency, accuracy, stability, and security metrics. It reveals performance trade-offs among leading LLMs and identifies security vulnerabilities and inconsistencies absent from synthetic benchmarks.

Strengths:
- Practical focus on real-world enterprise needs with a comprehensive five-metric evaluation framework
- Dataset derived from genuine enterprise interactions across multiple domains
- Thorough cross-model comparisons highlighting key deployment limitations
- Valuable jailbreak assessment revealing practical evaluations of SoTA LLMs

Weaknesses and Suggestions:
- Single-vendor data source (Aisera) potentially limiting generalizability
- Security evaluation restricted to jailbreak prompts, neglecting other critical vulnerabilities
- Narrow focus on workflow selection rather than multi-step reasoning or document retrieval
- Insufficient analysis of the causes behind observed performance instability

I still believe this paper should be accepted, though, because despite limitations in dataset diversity and security assessment scope, this paper makes a significant contribution to an application of LLM trust, and I believe it will be an interesting contribution to the conference.

---

### Official Review · Reviewer_GiNa · 2025-03-02
**Paper is easy to follow and is nicely written. The uploaded subset is quite small.**

**Rating:** 4
**Confidence:** 4

**Review:**

### Summary

This paper provides a benchmark for evaluating LLMs on workflow classification. Multiple metrics are provided including cost, latency, accuracy and stability, with security being used to evaluate jailbreak attacks.

### Strengths
1. Nicely written.
2. The figures are quite instructive.

### Weaknesses

1. The uploaded subset contains only 70 out of the 1793 conversations are reported. Though it is understandable that the complete dataset is under review, the given sample is hardly 4% of the overall dataset. I think this is not representative of the whole dataset.
2. The Latency metric does not make sense. Is the objective to check how much time the LLM is taking to respond? If so, then the latency of the user network also comes into play. Moreover, latency is a specification of the model and has nothing to do with the dataset.
3. It would have been interesting to see the actual prompts given for jailbreaking. But it is not available.

---

### Decision · Program_Chairs · 2025-03-02

Accept